# Experimental and Numerical Study of Pd/Ta and PdCu/Ta Composites for Thermocatalytic Hydrogen Permeation

**DOI:** 10.3390/membranes13010023

**Published:** 2022-12-24

**Authors:** Seungbo Ryu, Arash Badakhsh, Je Gyu Oh, Hyung Chul Ham, Hyuntae Sohn, Sung Pil Yoon, Sun Hee Choi

**Affiliations:** 1Center for Hydrogen Energy and Fuel Cell Research, Korea Institute of Science and Technology (KIST), Seoul 02792, Republic of Korea; 2PNDC, University of Strathclyde, Glasgow G68 0EF, UK; 3Department of Chemical Engineering, Inha University, Incheon 22212, Republic of Korea; 4Department of Energy and Environmental Engineering, KIST School, University of Science and Technology (UST), Seoul 02792, Republic of Korea

**Keywords:** hydrogen permeation, composite membrane, palladium, copper, separation, density functional theory

## Abstract

The development of stable and durable hydrogen (H_2_) separation technology is essential for the effective use of H_2_ energy. Thus, the use of H_2_ permeable membranes, made of palladium (Pd), has been extensively studied in the literature. However, Pd has considerable constraints in large-scale applications due to disadvantages such as very high cost and H_2_ embrittlement. To address these shortcomings, copper (Cu) and Pd were deposited on Ta to fabricate a composite H_2_ permeable membrane. To this end, first, Pd was deposited on a tantalum (Ta) support disk, yielding 7.4 × 10^−8^ mol_H_2__ m^−1^ s^−1^ Pa^−0.5^ of permeability. Second, a Cu–Pd alloy on a Ta support was synthesized via stepwise electroless plating and plasma sputtering to improve the durability of the membrane. The use of Cu is cost-effective compared with Pd, and the appropriate composition of the PdCu alloy is advantageous for long-term H_2_ permeation. Despite the lower H_2_ permeation of the PdCu/Ta membrane (than the Pd/Ta membrane), about two-fold temporal stability is achieved using the PdCu/Ta composite. The degradation process of the Ta support-based H_2_ permeable membrane is examined by SEM. Moreover, thermocatalytic H_2_ dissociation mechanisms on Pd and PdCu were investigated and are discussed numerically via a density functional theory study.

## 1. Introduction

The reliance on fossil fuels since the 1950s has caused a stable increase in greenhouse gas emissions and triggered the greenhouse gas effect, the main driver of global warming [1]. One of the approaches to mitigate global warming is to develop clean and renewable energy sources, such as wind, water, solar, and geothermal energy. However, sustainable and eco-friendly energy sources vary by region and lack the required infrastructure worldwide. Hydrogen (H_2_) is one of the most promising energy carriers, given its high gravimetric energy density and its clean conversion byproduct, namely, water [2]. The chemical energy stored in H_2_ can be directly converted into electricity by using fuel cells and/or heat by combustion [3]. To date, ~50% of all H_2_ is produced by natural gas reforming, whereas a mixture of gases is released as the product [4,5]. Utilization of other H_2_ carriers and sources such as ammonia [6], a liquid organic hydrogen carrier (LOHC) [7], methanol [8], and biomass [9] also requires exhaust purification before the end-use. Therefore, the separation technique is essential for obtaining pure H_2_ from a mixture consisting of H_2_, nitrogen (N_2_), carbon monoxide (CO), carbon dioxide (CO_2_), and water vapor (H_2_O) [10]. However, commercial methods such as pressure swing adsorption (PSA) are expensive and complicated while also suffering from low energy efficiency [11]. To alleviate this gap, H_2_ separation by using membranes has been developed as an alternative method to reduce the cost and complexity of H_2_ purification [12].

In the H_2_ separation membrane, Pd is an essential catalyst for both the dissociation and association of atoms in the H_2_ molecule, yielding a high purity of >99.99% in the permeate H_2_ [13]. H_2_ gas molecule permeation through Pd is arguably governed by the solution diffusion mechanism, in which H_2_ dissociates into H atoms at the Pd surface. Then, the H atoms diffuse via the membrane and associate into H_2_ molecules at the surface of the opposite side [12,14]. Thus, the separation via Pd is independent of the molecular size of the gas and the pore size of the membrane. Furthermore, the solubility of hydrogen in the bulk of Pd is high and rather temperature-independent compared with other catalytically active metals for hydrogen atom association/dissociation [15,16]. Given these advantages, a very thin Pd film with a defect-free surface can be used as a suitable H_2_ separation membrane. As Pd is too expensive as a monolithic membrane, especially in large-scale applications, many studies have tried to reduce the cost of Pd-based membranes. The usual method is to use a support material such as dense or porous metals, ceramics, or polymers [17,18,19,20,21,22] because this approach increases the mechanical strength of the membrane while decreasing the amount of Pd. Furthermore, body-centered cubic (BCC) metals are promising materials because they have a better ability to permeate H_2_ than face-centered cubic (FCC) metals [23]. Moreover, BCC metal is advantageous because it can permeate high-purity H_2_ with only a sub-microscopic Pd layer, while porous support (PSS) or porous nickel support (PNS) require a microscopic thickness of Pd [24,25].

In this study, we selected Ta as the support among the BCC metals. The Wolden’s group reported that a Pd/Ta composite membrane has strong chemical durability against oxidation [26]. In their study, oxidized Nb and V were observed after a H_2_ permeation test, while Ta was not degraded by oxygen. Generally, group 5 (VB) metals, including Ta, have been presented as being suitable for fabricating thin H_2_ permeation membranes due to their relatively low price, good mechanical stability, and good H_2_ permeation potential [23,27]. Besides the cost, one of the disadvantages of Pd as an H_2_ separation membrane is H_2_ embrittlement at low temperatures. When Pd is exposed to H_2_ below 300 °C, beta-hydride is formed, thereby increasing the membrane volume and also breaking the membrane surface [28]. This effect can be weakened at temperatures of >300 °C, but other adverse effects such as increased energy usage and the aggregation of Pd would remain. To overcome this, Pd has been commonly alloyed with other FCC metals, such as Ag and Cu [29,30]. The alloyed metal allowed the H_2_ separation membrane to operate at low temperatures as it prevents Pd from forming beta-hydride even at about 200 °C and metal aggregation. Previous studies have reported that the solubility of H_2_ is high when PdCu alloys have an FCC structure, and the diffusivity is high when the BCC structure exists [29]. As the permeability of the H_2_ separation membrane is calculated by multiplying the solubility and diffusivity, it is necessary to fabricate membranes with the corresponding dual-functionality. Therefore, it is also important to find a composition in which two structures (BCC and FCC) exist simultaneously, for which the optimum weight ratio is reported as Pd:Cu = 6:4 [29]. In summary, we have selected the combination of Cu and Pd for the following reasons: (i) to maintain the high sulfur poisoning resistance of the membrane [31,32], (ii) to leverage the cost-effectiveness of Cu (6–6.22 USD/kg [33]) in alleviating the costliness of Pd (48,226 USD/kg [33])-based membrane systems, and iii) to enhance the durability of Pd/Ta composite membranes as delineated as crucial in the literature [34,35].

In this study, the fabrication of a durable and cost-effective H_2_ separation membrane was attempted. To this end, costly Pd or a less-expensive PdCu alloy was deposited on a Ta support surface to prepare the composite membranes. The synthesis methods and the permeation performance of the as-prepared membranes were studied and compared in detail to determine the suitability of the fabrication technique and alloying in achieving the study’s goal. Moreover, the correlation between the fabrication method and the permeation performance was delineated. Finally, density functional theory (DFT) was also applied to evaluate and explain the experimental trends obtained herein.

The novelty of this study can be summarized as the following:Providing evidence that fabrication of nanometer-thick Pd and PdCu on a dense support is achievable via plasma sputtering.Analysis of temporal stability of Pd/Ta and PdCu/Ta membranes.

## 2. Methodology

### 2.1. Materials

Ta sheets with a thickness of 250 μm were purchased from Koralco Corporation (Gwangju, Republic of Korea). They were then wire-cut into disks with a diameter of one inch. Hydrochloric acid (HCl) and phosphoric acid (H_3_PO_4_) were supplied by Samchun Chemicals (Seoul, Republic of Korea). Tin chloride (SnCl_2_), palladium chloride (PdCl_2_), hydrazine, and ethylenediaminetetraacetic acid (EDTA) were purchased from Sigma-Aldrich (St. Louis, MO, USA). Tetraamminepalladium dichloride monohydrate (Pd(NH_3_)_4_Cl_2_·H_2_O) was purchased from Sigma-Aldrich (St. Louis, MO, USA). All chemicals were used as-received and without further purification unless stated otherwise.

### 2.2. Membrane Preparation

The preparation process is shown in Appendix A. The Pd layer was electroless-plated (ELP) on pre-treated Ta discs and tubes. The surface of the Ta support was polished in the following order. First, Ta supports were polished using 800, 1200, 1500, 4000, and 7000 grit sandpaper. Subsequently, as mentioned in the literature [17], the impurities on the membrane were removed using a basic solution, and organic substances were removed through acid treatment using stepwise immersion in HCl and H_3_PO_4_ aqueous solutions. The Pd ELP is based on a well-established metal–metal galvanic exchange technique [36]. To this end, the surface was activated by sequential dipping of the membrane in 1.0 g/L SnCl_2_ and 1.0 g/L PdCl_2_ aqueous solutions, each containing 0.01 M HCl as the stabilizing agent. This process was repeated three times, and each step was conducted for 5 min. Finally, the ELP was performed at a temperature of 60 °C in a Pd(NH_3_)_4_Cl_2_ bath following the details mentioned in [17].

For the PdCu/Ta membrane, the sample was prepared using ELP and plasma sputtering (SPT). First, Pd was plated using the method described above. Then, Cu was deposited on the Pd layer by using a magnetron sputtering system (Korea Vacuum Tech, Gimpo-si, Korea). Subsequently, the prepared membrane was treated at 480 °C for 1 h in H_2_ atmosphere for alloying. The co-sputtered (co-SPT) PdCu/Ta membrane was also prepared by using a plasma sputtering system to avoid the adverse effect of Sn residue from SnCl_2_ during the activation process. The sputtering conditions were 10 cm (distance between the target and substrate), 25 W (Pd, DC), 18–45 W (Cu, RF) in 20 NmL/min of Ar stream, 2 mTorr of working pressure, and deposition temperatures of the room (~20 °C, R.T.) and 400 °C (Pd). The PdCu/Ta membranes were also prepared with ELP Pd and SPT Cu, successively. After deposition, Pd and Cu were alloyed at 480 °C for 1 h in a H_2_ atmosphere.

### 2.3. Permeation Testing

The tests were conducted by using high-purity (99.999%) H_2_ gas for the permeation test and Ar gas for purging during the heating and cooling processes. The gas flow rate was adjusted using a thermal mass flow controller (MFC, Bronkhorst High-Tech BV, Ruurlo, Netherlands), while the pressure at the membrane terminals was adjusted using an electric pressure controller (EPC). The permeate gas flow rate was measured by a mass flow meter (MFM, Bronkhorst High-Tech BV, Ruurlo, Netherlands) and a bubble flow meter (BFM, Horiba, Kyoto, Japan). The measurement system is described in detail in Figure 1. The supplied gas was either passed through the H_2_ separation membrane in the reactor (permeate) or separated to escape the furnace (retentate). In the experiment, H_2_ permeability was measured at 400, 425, 450, 475, and 500 °C (ramp-up rate: 5 °C/min) and the pressure range of 1–5 bar. A temporal stability test for H_2_ permeation was performed at the 500 °C and 5 bar conditions until the membranes were broken.

### 2.4. Material Characterizations

The surface and cross-sectional images of the prepared membranes were retrieved through scanning electron microscopy (SEM, Inspect F-50, FEI Company, Hillsboro, OR, USA), and elemental mapping was performed using an energy dispersive spectrometry detector (EDS, AMETEK Inc., Berwyn, PA, USA). An accelerating voltage of 15 kV was used for SEM analysis unless stated otherwise. Before retrieving the cross-sectional images, the samples had been molded and cured in epoxy resin for one day. Besides these characterizations, the prepared membranes were characterized using an X-ray diffractometer (XRD, D’ Max 2500, Rigaku, Tokyo, Japan) to investigate the lattice of alloyed metals and Rutherford backscattering spectroscopy (RBS, National Electrostatics Corporation, Middleton, WI, USA) to determine the composition of the alloyed metal prepared by ELP and SPT, respectively. XRD scanning range was 10–90 deg with a step size of 0.02 (2θ).

### 2.5. DFT Modeling

Spin-polarized density functional theory (DFT) calculations with the aid of the Vienna Ab-initio Simulation Package (VASP, University of Vienna, Vienna, Austria) [37] were also conducted, and the exchange-correlation function was described using the Perdew Burke–Ernzerhof generalized gradient approximation (GGA) method [38]. The projector augmented wave (PAW) method was applied to substitute complicated ionic potentials caused by the interaction between the ion and electron cores [39]. A plane wave expansion with a cutoff energy of 400 eV was used to express the valence electrons. A 5 × 5 × 1 Monkhorst–Pack mesh k-point was utilized to determine the optimal geometries and total energy with sufficient accuracy [40].

Four different model Pd surfaces were prepared, including face-centered cubic (FCC) Pd (111), body-centered cubic (BCC) Pd (110), FCC PdCu (111), and BCC PdCu (110) with 50 at.% Pd to understand the H_2_ dissociation on the surface and the diffusion of atomic H into the membrane. Each slab was modeled using 2 × 2 six-layer supercells, with all the layers relaxed. Although Pd exists as the FCC crystal structure in the standard state, BCC Pd is considered for comparison against BCC PdCu. The quantified lattice parameters of FCC Pd, FCC PdCu, BCC Pd, and BCC PdCu were, 3.94 Å, 3.81 Å, 3.23 Å, and 2.99 Å, respectively. These estimates showed good agreement with the empirical estimates [*a*_Pd_ = 3.89 Å, *a*_(FCC Pd52Cu48)_ = 3.77 Å and *a*_(BCC Pd47Cu53)_ = 2.97 Å] (see Table 1). Moreover, the climbing image nudged elastic band (CI-NEB) method [41] was applied to quantify the energy barriers for H_2_ dissociation on the modeled Pd and PdCu surfaces. Six images between the initial and final adsorption geometries were generated for this purpose. Equation (1) was used to calculate the binding energy (*E*_bind_) of H_2_ (or H):(1)$$E_{\mathrm{bind}}=E_{H_{2}\left(\mathrm{or}\;H\right)/\mathrm{slab}}-\left(E_{\mathrm{slab}}+E_{H_{2}\left(\mathrm{or}\;H\right)}\right)$$
where $E_{H_{2}\left(\mathrm{or}\;H\right)/\mathrm{slab}}$, $E_{H_{2}\left(\mathrm{or}\;H\right)}$, and $E_{\mathrm{slab}}$ are the total energy of the H_2_ (or H)-adsorbed slab, gaseous H_2_ (or H), and pure slab systems, respectively.

## 3. Results and Discussion

### 3.1. Pd/Ta Membrane

In the Pd/Ta H_2_ separation membrane, the permeability was measured at 450–500 °C and 1–5 bar of pressure difference, as shown in Figure 2a. For these metal membranes, H_2_ permeates in the following order: adsorption, dissociation, volumetric diffusion, association, and desorption. Equation (2) establishes the permeate flux of these membranes based on Sievert’s law:(2)$$J=\frac{Q_{f}}{l}\left(P_{feed}^{n}-P_{perm}^{n}\right)$$
where *Q*_f_ represents the permeability of the membrane, *l* is the thickness of the membrane, *P*_feed_ is the pressure of the front part before permeation, and *P*_perm_ is the pressure of the latter part after permeation. In the H_2_ separation membranes, *n* = 0.5 if the permeation rate is determined by diffusion through the metal layer, while *n* = 1 if the rate is determined by the H_2_ dissociation/association reaction on the surface, and 0.5 < *n* < 1 if both apply [45]. It is generally thought that H_2_ penetration in a metal layer containing Pd follows the Sievert’s law, which means *n* = 0.5 [46]. In this case, the atmosphere was close to the ideal gas conditions, and the rate of H_2_ permeation was mainly conducted through the metal lattice. Figure 2 shows the permeability of the Pd/Ta composite membrane at various temperatures and pressures. The H_2_ permeability of the Pd/Ta membrane was measured as 16.18 cm^3^·cm^−2^·min^−1^. This illustrates a 7.4 × 10^−8^ mol_H_2__ m^−1^ s^−1^ Pa^−0.5^ permeability which is well within the range reported previously for Pd/Ta membranes [34]. At 500 °C, H_2_ permeated through the membrane at *n* = 0.5. However, *n* was close to one in the 475 °C and 450 °C experiments. This phenomenon could be driven by the degradation of the membrane surface with the increasing experimental time. In this experiment, H_2_ permeability was measured from high to low temperatures.

Figure 2b illustrates the comparison of the permeability of H_2_ separation membranes prepared by various methods at 500 °C and a pressure difference of 5 bar. In the H_2_ separation membrane, fabricated by ELP and SPT at 400 °C, the H_2_ permeability exhibited nearly similar values. The H_2_ separation membrane, deposited at room temperature, exhibited a significantly lower estimate than the other two membranes. This phenomenon is attributable to the difference in the density of the Pd surface.

In general, due to the sintering effect, the metal deposited at high temperatures is denser than the one sputtered at room temperature. When Pd is deposited on the substrate at a high temperature, Pd atoms easily move on the substrate, thereby forming a denser layer during sputtering deposition [47]. This was confirmed by the SEM images of the samples before the H_2_ permeation test, as shown in Figure 3. As seen, the membranes prepared by ELP and SPT at 400 °C have denser structures than the room-temperature SPT surface with many defects. The SPT Pd layer deposited at RT has a relatively larger size (~10 nm, thus, agreeing with the available literature [48,49]) and a larger number of pores than the SPT Pd deposited at 400 °C and ELP Pd. On top of that, almost no pores were identified on the surface.

Moreover, the morphology of the H_2_ separation membranes was changed after the permeation test for 10 h. The porous structure of the SPT samples was changed to a smoother surface due to sputtering at the operating temperature. Large pore islands were formed by the agglomeration of Pd on the surface. After long exposure to a high temperature, Pd becomes more aggregated, revealing the Ta support layer [34]. This degradation is a critical weakness in the H_2_ separation membranes during long-term use. To address this, PdCu alloys were also prepared and tested in this study.

Figure 4 shows the SEM images of the cross-sections of the ELP and SPT Pd/Ta membranes after the H_2_ permeation experiment. In contrast to Figure 1d (for Pd/Ta before the H_2_ permeation test), the delamination was identified on the cross-section of the membrane after the experiment. This finding confirms that separation occurred precisely at the interface between Ta and Pd. As the experiment was conducted at a high temperature (500 °C), the surface of Ta, directly exposed to H_2_, increases as Pd aggregates on the surface. Then, delamination occurs, given the difference in the H_2_ embrittlement between the two metals. Unlike Pd with generally low H_2_ embrittlement at high temperatures, Ta exhibits the opposite trend [23,50]. In addition, the Sn used in electroless plating can also affect delamination.

Furthermore, the Sn residue in the ELP method reduces the adhesion of Pd to the Ta substrate, thus, promoting the delamination of Pd (see Appendix A). This phenomenon prevents a uniform supply of hydrogen atoms to Ta, subsequently leading to a higher degree of delamination, which can ultimately reduce the stability of the H_2_ separation membrane in long-term operations. The delamination occurs in the membranes prepared by sputtering without Sn, but this delamination occurs at smaller scales and more sporadically. This indicates that the H_2_ embrittlement and Sn residue factors both affect the degradation of the metal-support-based H_2_ separation membrane.

### 3.2. PdCu/Ta Membrane

Besides the Pd/Ta membrane, a layer of PdCu alloys was deposited on Ta supports to mitigate the disadvantages of Pd, including the weak durability and high cost. PdCu/Ta membranes were prepared using two methods: (i) electroless-plated (ELP) Pd, followed by sputtered Cu (SPT), and (ii) co-sputtered PdCu (co-SPT) on the Ta disk. Figure 5 indicates that the formation of a uniform Pd-based coating is achievable on dense supports, which is a silicon wafer in this case. The thickness of the co-sputtered layer is measured to be 446.8 nm, and it is at least one order of magnitude lower than the Pd coating on the porous supports reported by previous studies [45]. The advantage of thinner Pd-based alloy could also be supported by the work of Ramachandran et al. [51], where they illustrated that H_2_ permeability is inversely correlated with Pd-based membrane thickness.

Figure 6 shows the characterization of the co-SPT alloy membrane by using XRD and RBS. X-ray diffractograms demonstrate that Pd and Cu exist as alloys and not as separate phases. Note that quantitative analysis of the alloys was possible by using RBS, which confirmed the alloy composition present in the membrane. The PdCu alloy has different metal lattices depending on the component ratio. The highest performance ratio is the 6:4 weight ratio (47:53 mol ratio), where both FCC and BCC structures exist simultaneously [29]. In general, the FCC lattice has high solubility, and the BCC lattice has high diffusivity. Figure 6b shows that the optimum alloying ratio has been achieved for the as-prepared PdCu(co-SPT)/Ta membrane.

Figure 7 shows the H_2_ permeability of PdCu/Ta membranes at various temperatures (400–500 °C) and pressure gradients (0.41–1.45 bar^0.5^). As expected, an increase in either temperature or pressure gradient results in higher H_2_ permeability. The Pd(ELP)Cu(SPT)/Ta membrane exhibited 9.66 cm^3^·cm^−^^2^·min^−1^ at 500 °C and a pressure difference of 5 bar, which is three times higher than that of the PdCu(co-SPT)/Ta membrane. If Pd and Cu are both deposited by ELP, the plated Pd will detach when Cu is plated. This occurs because Cu replaces the Pd atoms when plated under higher pH conditions than Pd plating. Previous studies have shown that Pd precursors can be used to activate the surface of supports to conduct Cu ELP, with an appropriate pH of 11 [52,53]. Moreover, in the case of co-sputtering Pd and Cu, some practical challenges emerge. Although the alloy composition can be easily controlled by co-sputtering, we identified a low H_2_ permeation problem. It has been likely driven by less dense structures at room temperature deposition (see Figure 7a). In contrast, the side effects of high-temperature deposition, such as the formation of unwanted alloy phases, could simultaneously emerge. Meanwhile, the method of sputtering Cu on an electroless-plated Pd could solve the problem above, revealing a stable high H_2_ permeability, as shown in Figure 7b.

Moreover, the maximum permeability obtained for the PdCu/Ta membrane was 4.4 × 10^−8^ mol_H_2__ m^−1^ s^−1^ Pa^−0.5^ which is more than the reports on PdCu membranes with permeabilities ranging from 0 to 2.75 × 10^−8^ mol_H_2__ m^−1^ s^−1^ Pa^−0.5^ [54]. The H_2_ permeability in the Pd(ELP)Cu(SPT)/Ta membrane was ~60% of that in the Pd/Ta membrane. However, it exhibits excellent durability at high temperatures, where the *n* value does not change with the experimental time. The constant value of *n* indicates that the rate of H_2_ molecule decomposition on the surface remains constant during the experiment. This phenomenon can be interpreted as a lower degree of surface degradation compared with the Pd/Ta membrane. This phenomenon can be confirmed by the SEM images of the Pd(ELP)Cu(SPT)/Ta membrane before and after the temporal stability test, as shown in Figure 8. As a result, the surface of the Pd(ELP)Cu(SPT)/Ta membrane has not degraded after the test, unlike the Pd/Ta membranes (see Figure 3). Only slight agglomeration by heat and no bare Ta surface islands were identified in the Pd(ELP)Cu(SPT)/Ta case. This indicates the absence of degradation due to the high temperature. This result is consistent with the constant n value in the H_2_ permeation test. As Ta does not directly interact with H_2_, the deterioration due to H_2_ embrittlement of Ta is considered to be averted.

Lastly, calculations assuming the cost of Pd at (48,226 $/kg [33]) and Cu at (6.11 $/kg [33]) show that Pd coating with the as-fabricated dimensions (~4 μm in thickness and 25.4 mm in diameter) costs ~1157.4 $/m^2^, whereas the as-proposed PdCu alloy costs ~57% less, well within the United States Department of Energy’s standard of <5400 $/m2 [55].

### 3.3. Arrhenius Plot

According to the van’t Hoff–Arrhenius equation, the relationship between H_2_ permeability and temperature can be expressed by Equation (3):(3)$$\ln P=P_{0}\exp\left(\frac{E_{a}}{RT}\right)$$
where *P* is the permeability (mol/m^2^ s Pa^0.5^), *P*_0_ is the pre-exponential coefficient, *E*_a_ is the apparent activation energy, *R* is the ideal gas constant (J/mol K), and *T* is the absolute temperature. The activation energies can be quantified from the slope of the ln*P*-1000/*T* graphs, as shown in Figure 9. The estimates are 20.2 kJ/mol for the Pd(ELP)/Ta disk, 26.4 kJ/mol for the Pd(ELP)Cu(SPT)/Ta disk, and 38.4 kJ/mol for the PdCu(co-SPT)/Ta disk.

The H_2_ decomposition reaction on the membrane surface is endothermic. The reaction on the Pd surface has lower activation energy than that on the PdCu alloy surface. These values are consistent with the experimental permeations, where the Pd disk has the fastest permeation rate and the lowest activation energy. This confirms that the H_2_ permeation through palladium is faster than that of the PdCu alloy. Moreover, the difference in the activation energy between the PdCu alloy membranes in the H_2_ permeability results was also identified; compared with Pd/Ta, the difference was 1.45 times and 2.15 times higher for Pd(ELP)Cu(SPT)/Ta and PdCu(Co-SPT)/Ta, respectively. This is in good agreement with the results in Figure 7, where the membrane made by successive ELP and SPT exhibited a significantly higher (up to 5.9 times) H_2_ permeability value than the membrane fabricated by co-SPT. It is considered that the non-dense alloy structure on the membrane surface, observed through SEM, acts as a constraint with regards to the H_2_ permeation and increases the activation energy. Godbole et al. used NiO thin films and concluded that the activation energy of the reaction could be increased as the surface roughness is increased [56]. In other words, the activation energies are different even with the same alloy.

### 3.4. H_2_ Permeation Modeling

For a deeper understanding of the H_2_ permeation difference between PdCu/ and Pd membranes, DFT calculations were performed on the reaction energy/barrier for H_2_ dissociation and the energy for H migration in FCC Pd (111), BCC Pd (110), FCC PdCu (111), and BCC PdCu (110) slabs. Figure 10 and Table 2 display the optimized adsorption configuration and binding energy (indicated by E_bind_) for one H_2_ molecule and two H atoms on the four membrane models. On the surface of each membrane, H_2_ was adsorbed onto the top site and then dissociated into two hollow sites: two fcc sites (fcc-fcc) for FCC Pd (111) (Figure 10a), one fcc1 site and one hcp1 site (associated with two Pd and one Cu atom) for FCC PdCu (111) (Figure 10b), two hollow sites for BCC Pd (110) (Figure 10c), and two hollow1 sites (associated with two Pd and one Cu atom) for BCC PdCu (110) (Figure 10d). As shown in Table 2, the activation energies for H_2_ dissociation into two H atoms on the FCC Pd (111) and BCC Pd (110) surfaces were lower than those on the FCC PdCu (111) and BCC PdCu (110) surfaces. This finding indicates that the pure Pd membrane has higher catalytic activity in the initial formation of H atoms on the surface than in the PdCu case.

Figure 11 shows the change in H binding energy as an H atom diffuses through the six-layer slabs of FCC Pd (111), FCC PdCu (111), BCC Pd (110), and BCC PdCu (110). It was found that the maximum decrease in H binding energy was by 0.57 eV [FCC Pd (111)], 0.65 eV [FCC PdCu (111)], 0.45 eV [BCC Pd (110)], and 0.60 eV [BCC PdCu (110)] during the migration process from the top surface layer to the inside layer of slab. This suggests that the Pd membrane is more conductive than the PdCu membrane for H_2_ permeation from the energy point of view, which is also supported by the work of Huang and Chen [57]. This finding resonates well with the experimental observations.

### 3.5. Comparative Temporal Stability Tests

A temporal H_2_ permeation test was performed using the sample with the highest permeability, namely Pd(ELP)/Ta and Pd(ELP)Cu(SPT)/Ta H_2_ separation membranes. H_2_ permeability was continuously measured, while maintaining the hardest conditions of the H_2_ permeation test conditions, namely, 500 °C and a pressure difference of 5 bar. Figure 12 shows the results of the H_2_ permeation for 14 h. In this graph (red line), H_2_ permeability decreased rapidly with time for Pd(ELP)/Ta. This trend confirms that Pd deteriorates quickly at high temperatures. Notably, after ~8 h, the H_2_ permeability converged to 0. In other words, it takes ~8 h to lose the H_2_ permeation ability as Pd aggregates exposing the Ta substrate to H_2_, which in turn reduces the catalytic activity of the Pd layer and increases the susceptibility of the exposed Ta to mechanical breakage due to H_2_ embrittlement. On the other hand, the trend of the temporal permeation test result for the Pd(ELP)Cu(SPT)/Ta (black line) membrane is noticeably different. Even though the overall trend was decreasing, similar to Pd(ELP)/Ta, the decreasing trend was considerably slower. This finding is in line with the surface degradation of the Pd(ELP)Cu(SPT)/Ta, which was delayed in the H_2_ permeation experiment. As a result, higher permeability of H_2_ was maintained for twice as long as that of the Pd(ELP)/Ta membrane.

## 4. Conclusions

In this study, Pd/Ta and PdCu/Ta composite membranes were prepared using different synthesis methods. The morphology (as-prepared and post-mortem), H_2_ permeability, and durability of the samples were empirically evaluated. The effect of Cu alloying on the catalytic performance and activation energy of the composite membrane were theoretically studied and discussed. Our study reports several important findings summarized as the following:

First, compared with sputtering, electroless plating yields a more uniform and defect-less Pd coating on the Ta substrate. Moreover, a higher temperature is more conducive for a lower degree of porosity in sputtering. Second, Pd/Ta prepared by electroless plating and high-temperature (400 °C) sputtering exhibited a similar trend in the H_2_ permeation rate. Third, the activation energy, calculated for Pd/Ta, was lower than that for PdCu/Ta, indicating a lower catalytic activity as expected by the lesser amount of Pd in the alloy. PdCu/Ta, prepared by electroless Pd plating and sputtered Cu, revealed lower activation energy than co-sputtered PdCu. The DFT modeling results confirmed the lower catalytic activity of PdCu/Ta compared with that of the Pd/Ta membrane. Moreover, unlike using porous supports, developing a Pd or PdCu film with a sub-micron thickness (~500 nm) on a dense Ta substrate is possible while using the sputtering (former) or co-sputtering (latter) techniques. Ultimately, the Pd layer on the Pd/Ta membrane agglomerated into a honeycomb shape was experimentally demonstrated. As a result, Ta decelerates the decomposition of H_2_ molecules on the surface thanks to the lower catalytic activity. At the same time, the permeability of PdCu/Ta was maintained twice as long as that of Pd/Ta. Though the temporal stability test shown here is for 14 h due to the compromised mechanical stability of embrittled membranes, one can apply the findings of this study, in terms of the materials and synthesis method, in different membrane geometries and configurations, e.g., a tubular membrane reactor, to achieve a longer service life. Finally, we can conclude that alloying with Cu can open a window to long-term H_2_ operation compared with the bare Pd membrane. This greatly contributes to the development of cost-effective and durable membranes for thermocatalytic H_2_ separation and purification.

## Figures and Tables

**Figure 1 membranes-13-00023-f001:**
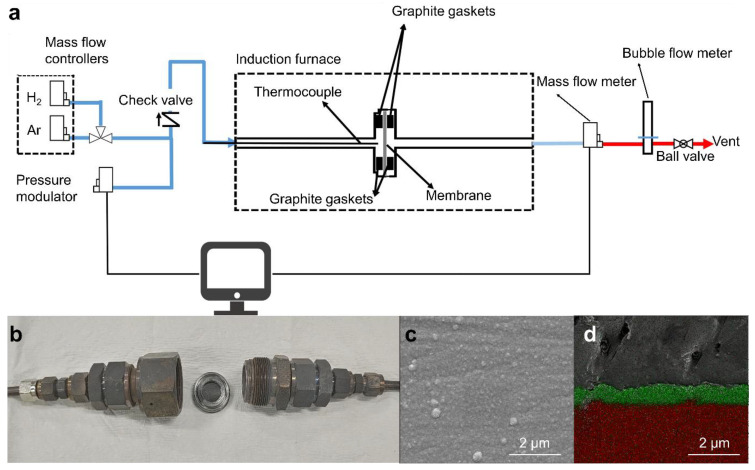
(**a**) Schematic diagram and photograph of H_2_ permeation system, (**b**) reactor and its parts, (**c**) SEM image of the surface, and (**d**) EDS mapping of cross-section of Pd on Ta support: green—Pd, and red—Ta.

**Figure 2 membranes-13-00023-f002:**
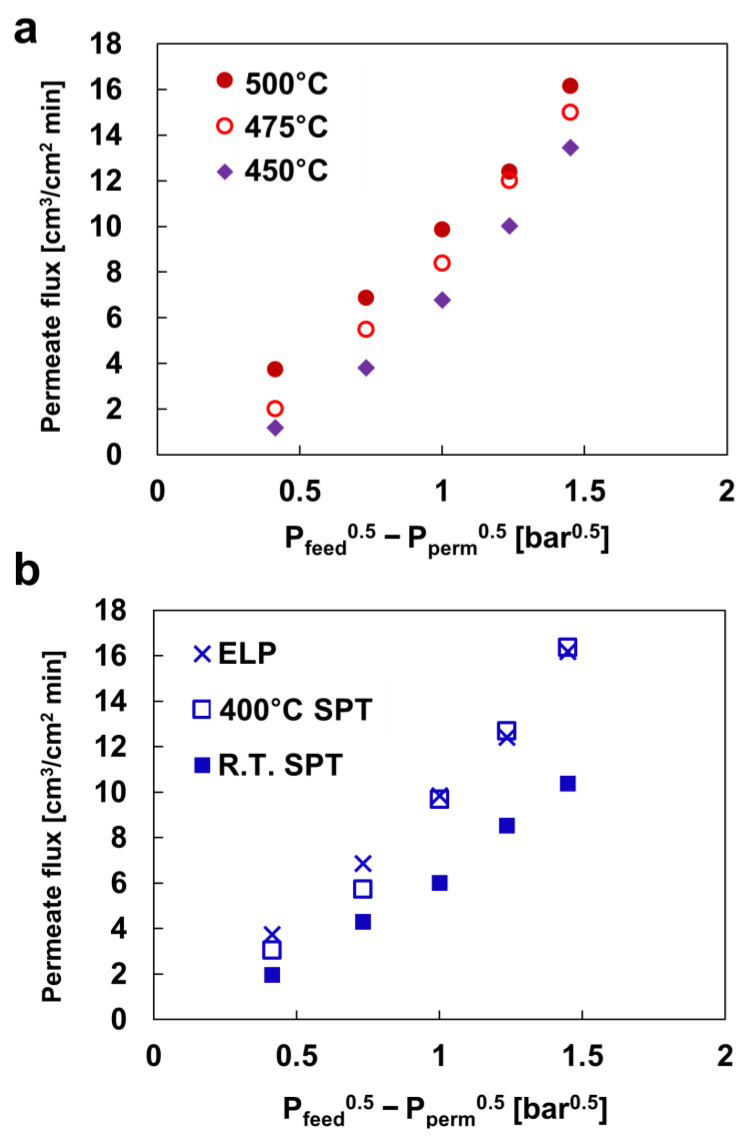
H_2_ permeability of (**a**) electroless-plated (ELP) Pd/Ta membrane and (**b**) the membranes prepared by different methods at 500 °C and 5 bar.

**Figure 3 membranes-13-00023-f003:**
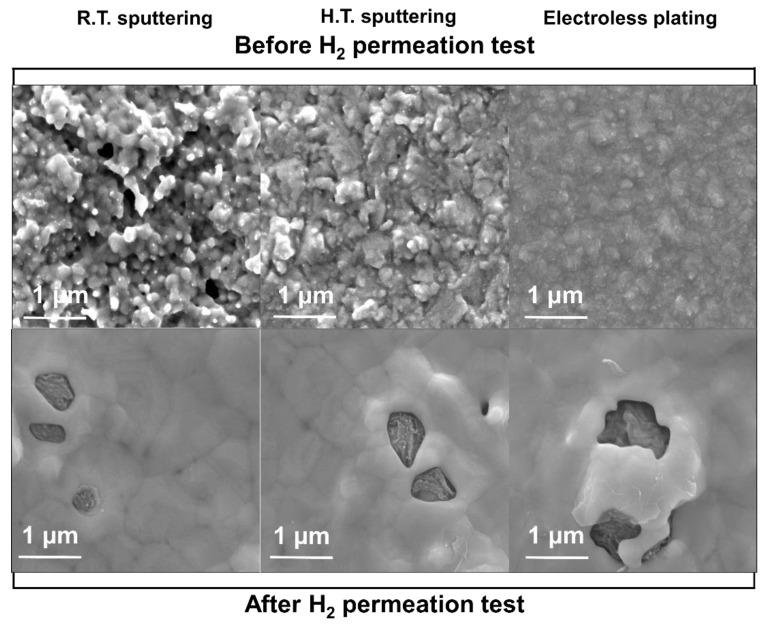
SEM surface micrographs (50,000 magnification) of the Pd/Ta membrane prepared by sputtering (SPT) at room temperature (R.T.) and high temperature (H.T.: 400 °C), and electroless plating (ELP), before and after the H_2_ permeation test at 500 °C.

**Figure 4 membranes-13-00023-f004:**
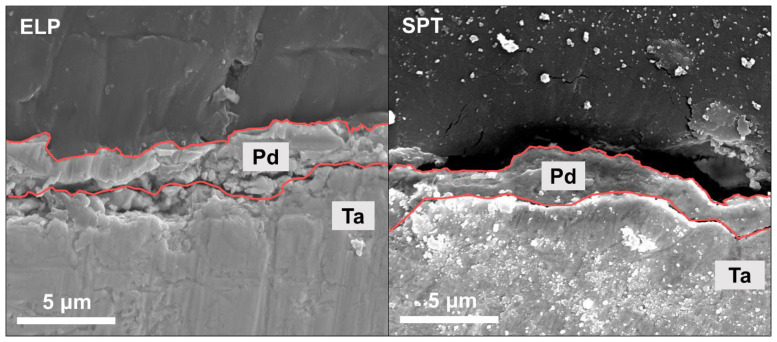
Cross-section SEM image of ELP and SPT Pd/Ta H_2_ separation membrane after the permeation test at 500 °C; Red lines mark the interfaces of Pd layer with Ta support and resin embedding.

**Figure 5 membranes-13-00023-f005:**
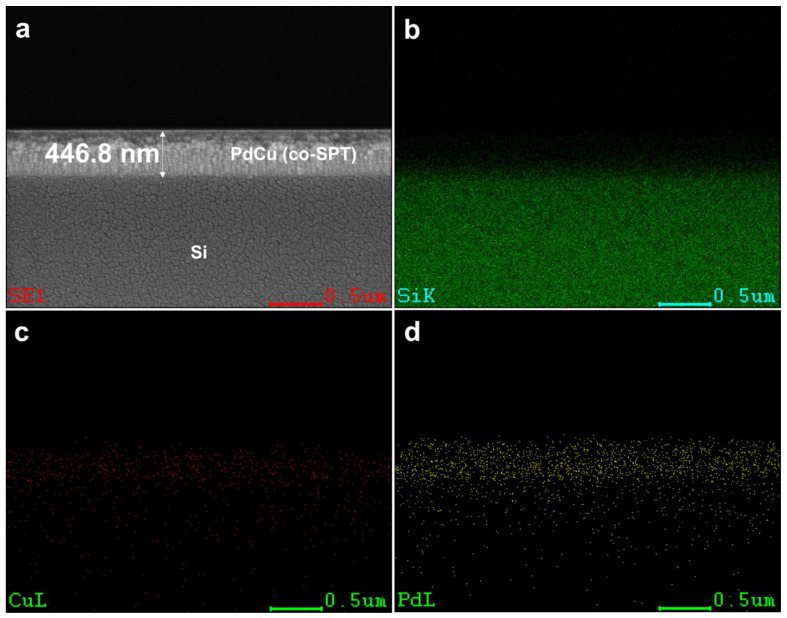
Cross-section (**a**) SEM image of PdCu(co-SPT)/Si, and EDS mapping of (**b**) Si, (**c**) Cu, and (**d**) Pd.

**Figure 6 membranes-13-00023-f006:**
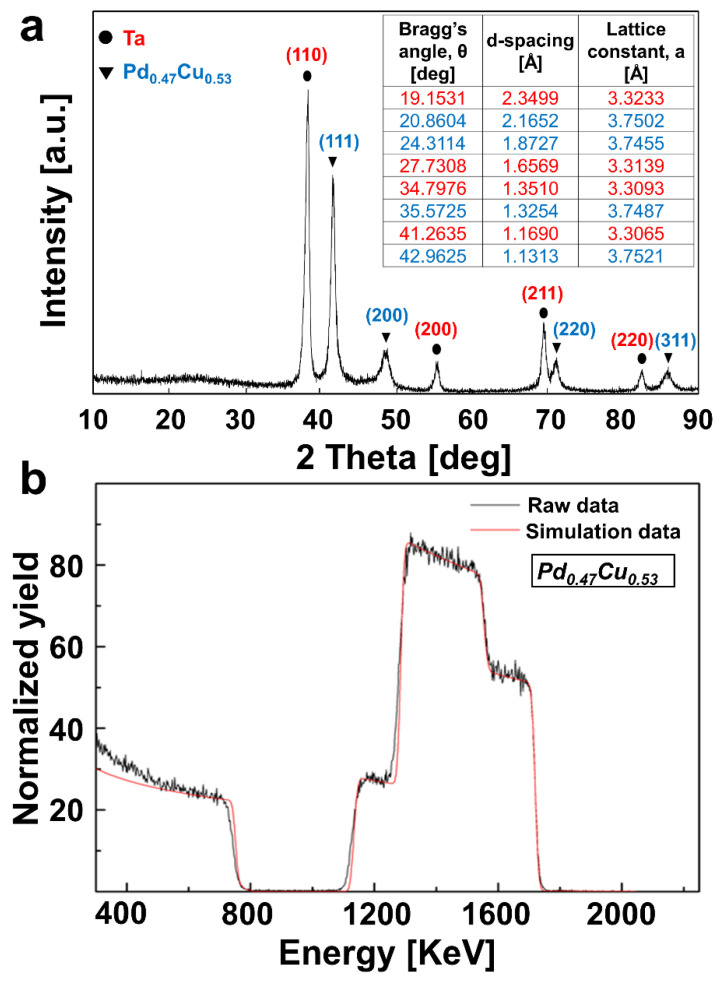
PdCu(co-SPT)/Ta characterization by (**a**) XRD and (**b**) RBS.

**Figure 7 membranes-13-00023-f007:**
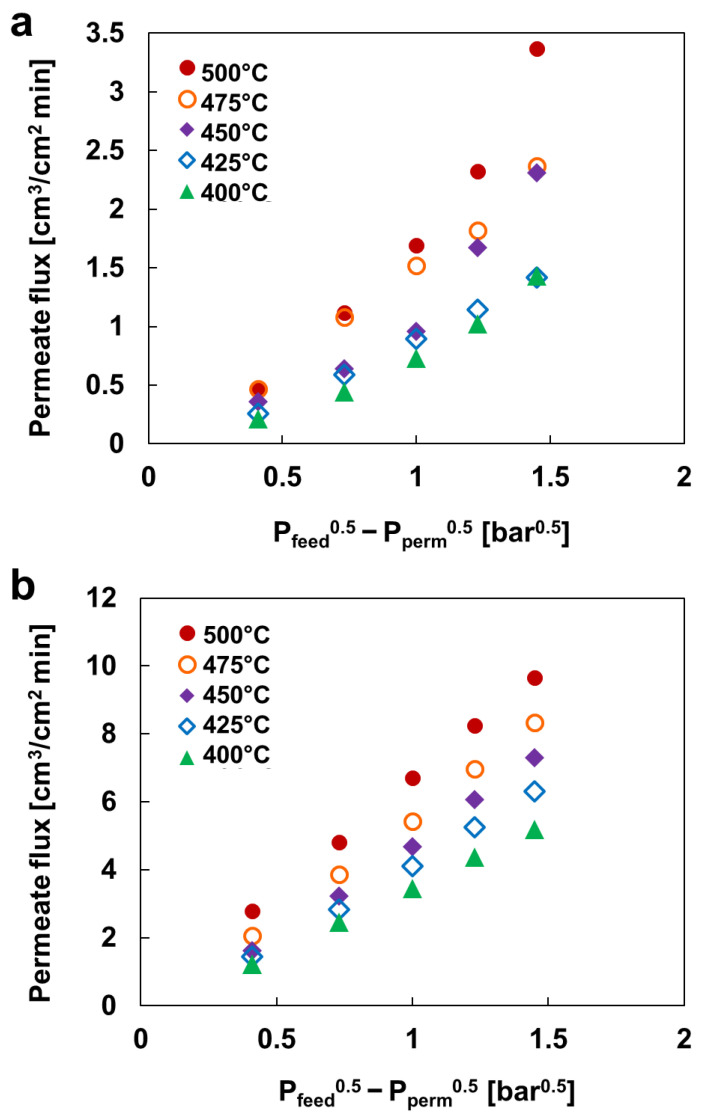
H_2_ permeability of (**a**) PdCu(co-SPT)/Ta and (**b**) Pd(ELP)Cu(SPT)/Ta membranes.

**Figure 8 membranes-13-00023-f008:**
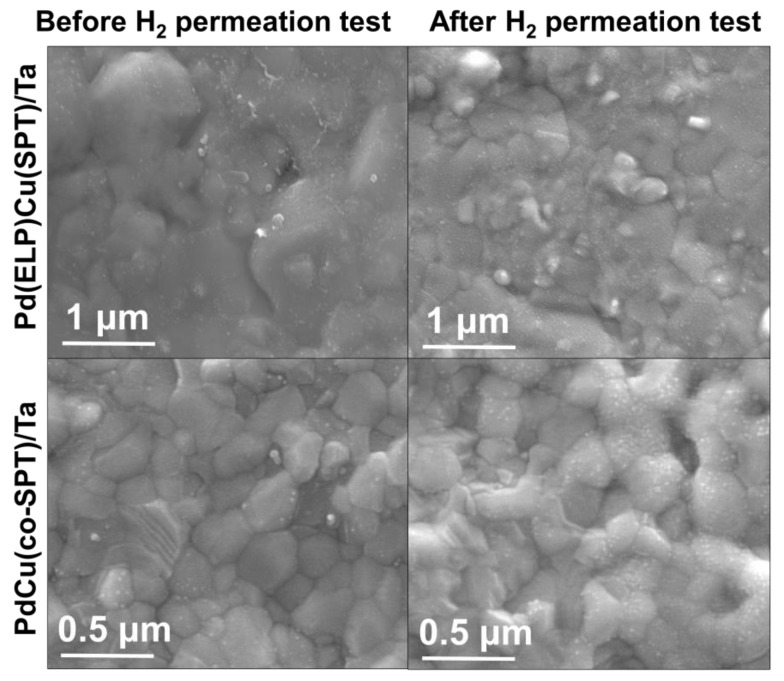
SEM (10 kV) images of surface of Pd(ELP)Cu(SPT)/Ta membranes before and after the H_2_ permeation test for 10 h.

**Figure 9 membranes-13-00023-f009:**
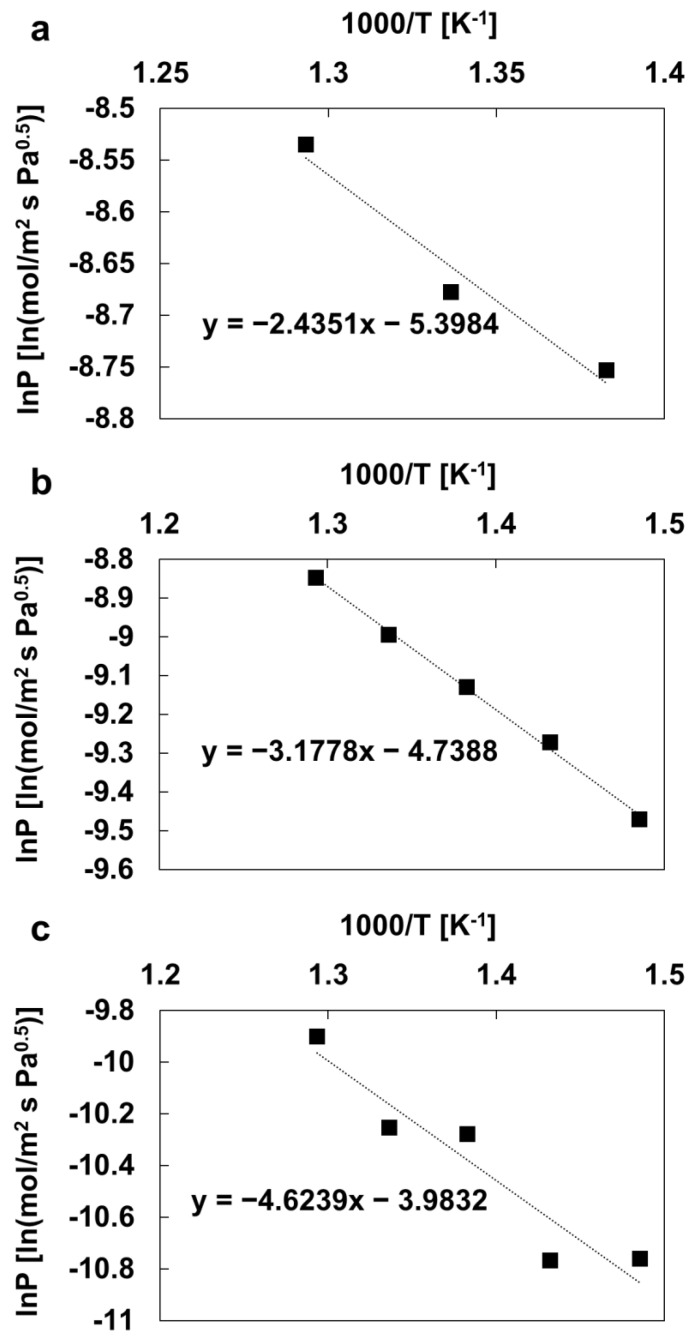
lnP vs. 1000/T of H_2_ permeation test for (**a**) the Pd(ELP)/Ta disk, (**b**) the Pd(ELP)Cu(SPT)/Ta disk, and (**c**) the PdCu(co-SPT)/Ta membranes.

**Figure 10 membranes-13-00023-f010:**
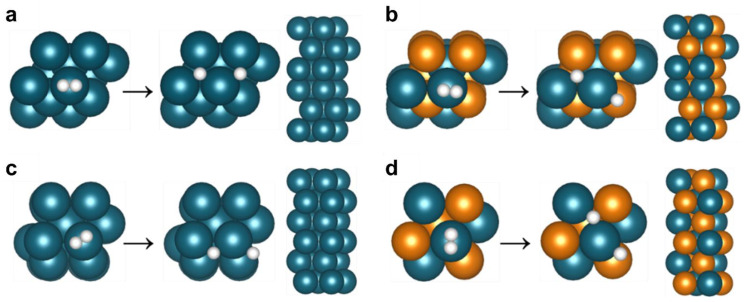
Top view for H_2_ dissociation and side view of slab models; (**a**) FCC Pd (111), (**b**) FCC PdCu (111), (**c**) BCC Pd (110), and (**d**) BCC PdCu (110). The teal, orange, and white balls denote the Pd, Cu, and H atoms, respectively.

**Figure 11 membranes-13-00023-f011:**
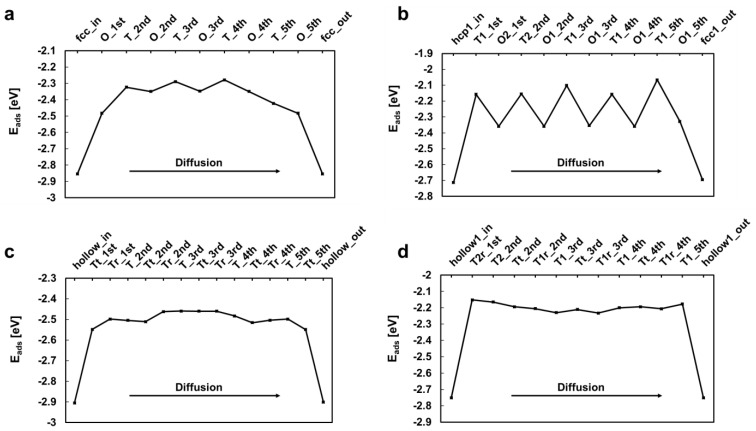
Variations of H binding energy as an H atom migrates through the Pd and PdCu 6-layer slabs. (**a**) FCC Pd (111); octahedral and tetrahedral sites are denoted as O and T, respectively. (**b**) FCC PdCu (111); octahedral sites associated with four Pd/two Cu atoms and two Pd/four Cu atoms are referred to as O1 and O2, respectively. Tetrahedral sites surrounded by two Pd/one Cu atoms and one Pd/two Cu atoms are denoted as T1 and T2, respectively. (**c**) BCC Pd (110); tetrahedral sites are denoted as Tt, Tr, and T along diffusion path. (**d**) BCC PdCu (110); tetrahedral sites associated with two Pd/two Cu atoms are referred to Tr, T, and Tt along the diffusion path.

**Figure 12 membranes-13-00023-f012:**
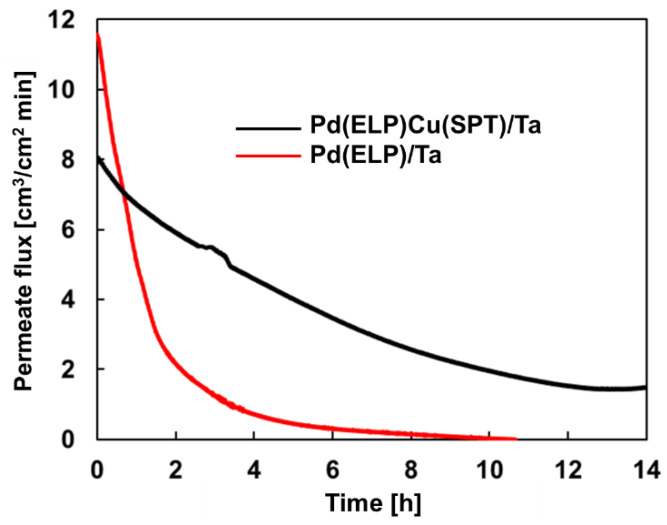
The comparative temporal stability test of Pd(ELP)/Ta and Pd(ELP)Cu(SPT)/Ta membranes.

**Table 1 membranes-13-00023-t001:** Estimates of lattice constants and comparison with experimental values in [42,43,44] for bulk FCC Pd, bulk FCC PdCu, bulk BCC Pd, bulk BCC PdCu, and bulk BCC Ta.

System	Alloy Composition [at.% Pd]	Lattice Constants [Å]	Experimental Lattice Constants [Å]	Experimental Alloy Composition [at.% Pd]
FCC pure Pd	100	3.94	3.89	100
FCC PdCu	50	3.81	3.77	52
BCC pure Pd	100	3.23	N/A	N/A
BCC PdCu	50	2.99	2.97	47
BCC pure Ta	0	3.31	3.31	0

**Table 2 membranes-13-00023-t002:** Calculated adsorption energies of H_2_ and activation energies for H_2_ dissociation reaction on FCC Pd (111), FCC PdCu (111), BCC Pd (110), and BCC PdCu (110) slabs.

System	H_2_ Adsorption Site	H_2_ Adsorption Energy [eV]	2H Adsorption Site	2H Adsorption Energy [eV]	Activation Energy [eV]
FCC Pd (111)	top	−0.23	fcc–fcc	−5.65	0.02
FCC PdCu (111)	top	−0.22	fcc1–hcp1	−5.35	0.10
BCC Pd (110)	top	−0.41	hollow–hollow	−5.83	0.06
BCC PdCu (110)	top	−0.27	hollow1–hollow1	−5.43	0.17

## Data Availability

The data presented in this study are available on request from the corresponding author.

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
