# Peer review of "Experimental and Numerical Study of Pd/Ta and PdCu/Ta Composites for Thermocatalytic Hydrogen Permeation"

_membranes, 2022, doi:10.3390/membranes13010023_

Round 1

Reviewer 1 Report

This study reports the preparation of PdCu/Ta membranes that can be utilized for the study of thermocatalytic hydrogen permeation. Although the article is almost well-managed, some revisions are needed before publication acceptance in the ‘’Membranes’’.

Comments for the authors:

  1. All the figures and tables should be cited in the manuscript.
  2. Some figures include several parts. To avoid confusion the parts should be coded and the related captions should encompass the details.
  3. The considered part of the SEM images can be highlighted using colorful lines.
  4. The abstract can be summarized by the obtained results.
  5. The novelty of this study should be stated in the Introduction.
  6. The results should be compared with the literature. The present form is like a scientific report that its achievements are not highlighted.
  7. The conclusion is too long and includes literature. Some parts can take place in the Results and discussion.
  8. The article's language should be altered thoroughly.

Reviewer 2 Report

Manuscript ID: membranes-2094909

Reviewer comments.

Ryu et al., have conducted a study entitled, “Experimental and numerical study of Pd/Ta and PdCu/Ta composites for thermocatalytic hydrogen permeation”, where the authors attempted to fabricate a durable and cost-effective H2 separation membrane. Both costly Pd and less-expensive PdCu alloy were deposited on a Ta support surface to prepare the composite membranes. The synthesis methods and the permeation performance of as-prepared membranes were studied and compared in detail to determine the suitability of the fabrication technique and alloying in achieving the study's goal. Moreover, the correlation between the fabrication method and the permeation performance was delineated. Finally, density functional theory (DFT) was also applied to evaluate and explain the experimental trends obtained. It is an interesting piece of research that has applications in the fields of thermocatalytic hydrogen permeation. I would recommend the manuscript be published in MDPI Membranes with the following criticisms addressed.

1-      The complete details of all the materials including "purchased from" and "used with/without further purification" etc should be mentioned in section 2.1

2-      The methodology seems interesting to me in the section 2.2. I would suggest the author to add a schematic diagram of the process for others to easily understand and follow!

3-      Page # 4, Fig. 1 caption should be corrected. Fig. 1(c) is SEM image and it should be mentioned in the caption. Fig. 1(d), is it a EDX image or simple SEM cross-sectional image? if simple cross-section image, how did author manage to get the colour of Pd and Ta?

4-      In the Section 2.4, please state the operating voltage of SEM.

5-      In the Section 2.4, please state the scanning angle range and step-size of the XRD.

6-      From Fig. 6 (a), XRD should present the information regarding lattice parameters, d-spacing and etc. in a tabular form!

7-      All figures except Fig. 10 (DFT) are presented in poor resolution and a professional software such as Origin etc. should be used to plot the graphs and be presented professionally in high-resolution.

8-       There are many errors those are found throughout the manuscript, few of which I have mentioned below:

Page # 3, Section 2.3, line # 6 "....Error! Reference....", Page # 5, line # 2 "....Error! Reference....", Page # 5, Section 3.1, line # 2 "....Error! Reference...."., Page # 5, Section 3.1, line # 14 "....Error! Reference....". Page # 5, Section 3.1, line # 21 "....Error! Reference....", Page # 6, , line # 7 "....Error! Reference....". Page # 7, line # 1 and 2 "....Error! Reference....". Page # 8, Section 3.2, line # 5 "....Error! Reference....",  Page # 9, line # 1 and 8 "....Error! Reference...."., Page # 11, line # 8 and 10 "....Error! Reference....". Please check these all references & errors and many more throughout the manuscript.

Author Response

Please see the file attached.

Reviewer 3 Report

1. a cutoff energy of 400 eV is used, whether is big enough?

2. What is about mechanism for thermocatalytic hydrogen permeation?

3. how is about the temporal stability of the samples?

4. The calculated Activation energy is very small, why?

5. The data should be compared with the references.

Author Response

Please see the file attached.
